# Graphene-Enhanced Surface Plasmon Resonance Liquid Refractive Index Sensor Based on Photonic Crystal Fiber

**DOI:** 10.3390/s19173666

**Published:** 2019-08-23

**Authors:** Bin Li, Tonglei Cheng, Junxin Chen, Xin Yan

**Affiliations:** College of Information Science and Engineering, State Key Laboratory of Synthetical Automation for Process Industries, Northeastern University, Shenyang 110819, China

**Keywords:** photonic crystal fiber, surface plasmon resonance, graphene, refractive index

## Abstract

A surface plasmon resonance (SPR) liquid refractive index sensor based on photonic crystal fiber (PCF) is proposed. The PCF is made of the exposed core structure, and the gold film is formed by electron beam evaporation within its defects. The sensitivity of the sensor is improved by coating graphene on the surface of the gold film. The experimental results show that the sensitivity of the sensor is increased by 390 nm/RIU after the introduction of graphene, and finally to 2290 nm/RIU. The experiment and simulation have a good consistency. Significantly, the sensor can be reused, and the measurement accuracy can be maintained.

## 1. Introduction

Since the first report of optical fiber surface SPR sensor, many scholars have devoted themselves to the research of biosensors [1]. When it comes to a biosensor, the refractive index detection of a liquid environment is indispensable. Common types of optical fiber SPR sensors have their own advantages and disadvantages. For example, the geometrically modified optical fiber SPR sensor for tapering, polishing and bending optical fibers reduces the mechanical strength despite its high sensitivity [2]. Fiber Bragg Grating SPR sensor ensures the integrity and mechanical strength of the fiber. However, its sensitivity is relatively low, and the grating requirements are high [3]. Photonic crystal fiber SPR sensor can not only adjust sensitivity through flexible structure design, but also has relatively high mechanical strength and high reliability in practical application [4].

Sensitivity is one of the most important performance parameters of sensors, which affects the accuracy and resolution of sensor detection. The sensitivity of the optical fiber SPR sensor can be improved by special treatment of the metal film, such as coating functional materials or specific materials. Since the sensor is approaching the upper limit of its sensitivity when the SPR is produced by the metal film, we continue to have difficulty in improving the performance of the sensor. For example, simply increasing the thickness of the metal film increases the cost or broadens the spectrum. Despite recent reports, the photonic crystal fiber (PCF) sensor designed by Islam et al. can obtain a spectral sensitivity of 25,000 nm/RIU with a gold SPR [5]. This is a very high sensitivity for sensors. But the production of the PCF is challenging. The significance of the functional coating layer is not to replace the metal film, but to increase the performance limit of the sensor. It also allows sensors to be particularly sensitive to some molecules to achieve one-to-one detection [6,7,8]. As a plasma material, graphene has some metal properties [9]. If graphene is deposited on the surface of the metal film, when it is in contact with a liquid environment, efficient charge transfer can make the interface between metal and graphene produce strong coupling and enhance the evanescent field of optical fiber, which can lead to the enhancement of SPR [10]. Optical fiber SPR sensing technology can determine the refractive index of liquid through the change of resonance wavelength. Therefore, this phenomenon of graphene can change the resonance wavelength and intensity of the sensor, and affect the performance of the sensor.

Graphene as a coating must be in contact with the detection liquid to be effective. Therefore, many types of PCF structures have been proposed by some scholars. Rifat et al. proposed that the gold film be covered in the specific pore inside PCF, and then graphene is coated on the surface of the gold film [11]. Using the wavelength interrogation method, a maximum refractive index (RI) sensitivity of 3000 nm/RIU in the sensing range of 1.46–1.49 is achieved. Yang et al. obtained the spectral sensitivity of 2520 nm/RIU by using alternating air holes coated with graphene-silver bimetallic layers as analyte channels [12]. But their approach may not be possible with current technology. Because how to deposit graphene in the air hole is a serious problem. Therefore, more people choose to coat metal and graphene on the outside of PCF. Liu et al. first plated gold on the outside of PCF, and then coated the gold film with graphene of 20 nm. In this way, they obtained the maximum spectral sensitivity of 7500 nm/RIU [13]. There are also problems with their designs, such as the fact that optical fibers are too thin to be easily fabricated. To solve the above problems, a liquid refractive index sensor based on SPR using photonic crystal fiber with an exposed core structure is proposed in this paper. We deposited controllable thickness gold film by electron beam evaporation in the depression, and then coated the gold film with graphene. The results of refractive index test for different mass fractions of glucose solution show that graphene enhances SPR and improves the spectral sensitivity of the sensor.

## 2. Design and Theory

The proposed sensor model structure is shown in Figure 1. There are two air holes with the diameter set to d = 12 μm and the spacing of air holes is set to Λ = 17 μm. In order to fabricate this kind of optical fiber, we have fabricated three different sized air holes in a glass rod with a diameter of 20 mm by ultrasonic processing. The larger air holes above are cut by a cutting machine to create a notch, and the optical fiber is fabricated by a fiber drawing tower. Although the PCF is our self-made optical fiber, we find that Kostecki et al. of Adelaide University had already designed a better optical fiber with the same structure [14]. The existence of air holes can restrict the propagation of light in the core. The core of the optical fiber refers to the position above the air hole and near the notch. We assume that light will travel in this area. But the fact is that it may leak into the optical fiber cladding. This is mainly due to our failure to control the size of air holes in the process of optical fiber drawing, which reduces the limitation of light. There are gold film and graphene in the optical fiber notch above the air hole, and the measured liquid covers the graphene. The exposed fiber core design makes light more easily accessible to gold film, graphene and measured liquids.

Through software simulation, we can get the effective refractive index and confinement loss of PCF, etc. In order to analyze the performance of the sensor. The existence of scattering boundary conditions and perfectly matched layer (PML) can be used to absorb the energy of outward radiation [15]. In the experiment, PCF should be immersed in the tested solution. The refractive index of liquid can be obtained by analyzing the transmission spectrum received by the spectrometer.

Since the refractive index of Silica varies with the wavelength of the incident light, it is necessary to use Sellmeier equation of Silica. The refractive index of Silica can be expressed by Equation (1) [16]:(1)$$n\left(\lambda \right)=\sqrt{1+\overset{m}{\underset{i=1}{\sum }}\frac{B_{i}\lambda^{2}}{\lambda^{2}-\lambda_{i}}}$$where *λ* is wavelength in μm, *m* = 3. *B*_1_, *B*_2_, *B*_3_ are 0.6961663, 0.407926 and 0.8974794, respectively. *λ*_1_, *λ*_2_ and *λ*_3_ are 4.67914826 × 10^−3^ μm^2^, 1.35120631 × 10^−2^ μm^2^ and 97.9340025 μm^2^, respectively. Which permittivity of Au is expressed by the Drude-Lorentz model, Equation (2) [17]:(2)$$\varepsilon_{m}=\varepsilon_{\infty }-\frac{\omega_{D}^{2}}{\omega \left(\omega +j\gamma_{D}\right)}-\frac{\infty \varepsilon \omega \Omega_{L}^{2}}{\left(\omega^{2}-\Omega_{L}^{2})+j\Gamma_{L}\omega \right)}$$where $\varepsilon_{\infty }=5.9673$ is the permittivity in high frequency. $\gamma_{D}$ is damping frequency. Δε=1.09 is the weighting factor. $\omega_{D}$ is the plasma frequency. ω=2πc/λ is the angular frequency. According to the reference, *c* is the velocity of light, $\omega_{D}/2\pi =\;2113.6\;\mathrm{THz}$ and $\gamma_{D}/2\pi =15.92\;\mathrm{THz}\;$ The frequency and spectral width of the Lorentz oscillator are $\Omega_{L}$ and $\Gamma_{L},$ respectively, $\;\Omega_{L}/2\pi =650.07\;\mathrm{THz},$ and Confinement loss is expressed by Equation (3):(3)$$L=8.686\times \frac{2\pi }{\lambda }Im\left[n_{eff}\right]\times 10^{4}\left(\mathrm{dB}/cm\right)$$

*Im*[*n_eff_*] is the imaginary part of the refractive index of the fundamental mode. Usually, we can get the spectral sensitivity (*S_λ_*) of the sensor Equation (4) [18]:(4)$$S_{\lambda }=\frac{{\Delta \lambda }_{peak}}{\Delta n}\left(\mathrm{nm}/\mathrm{RIU}\right)$$

Δ*λ_peak_* is used to define the displacement of resonance wavelength. Δ*n* is employed for changes in the refractive index.

According to the work of Nair [19] et al., we know that the complex refractive index of graphene can satisfy Equation (5) at lower layers. Therefore, considering the number of graphene layers used, we also set the same *n*′ expression of the complex refractive index of graphene in the simulation:(5)$$n^{\prime }=3+i\frac{c}{3}\lambda$$

## 3. Results

By analyzing the results of numerical simulation, as shown by the black curve in Figure 2, we find that the real part of the effective refractive index of the fundamental core mode decreases with the increase of the incident light wavelength. The red curve shows that the imaginary part of the effective refractive index produces a sharp peak with the change of wavelength. This is due to the formation of SPR on the surface of the gold film and the realization of phase matching conditions. The surface plasmon wave (SPW) resonates with the P-polarized light of the incident light, which leads to the increase of the confinement loss of the fundamental mode. When the refractive index of the measured liquid is 1.33, they resonate at 586 nm. Phase matching results in the transfer of fundamental mode energy to SPR mode. When the coupling strength is maximum, the confinement loss is 30 dB/cm. The resonant wavelength is affected by the structure of PCF and the refractive index of the liquid. Moreover, according to the refractive index of the liquid to be measured, the position of the resonance wavelength will also move along with it. Therefore, the refractive index can be detected by analyzing the resonant wavelength corresponding to different refractive index.

For the optical fiber sensor using SPR, the thickness of metal material has a great influence on its performance. In this paper, we mainly analyze the influence of the thickness of the gold film on the confinement loss of PCF and the resonance wavelength of SPR. As shown in Figure 3a, we simulate and compare the thickness of 30, 35 and 40 nm gold films. It can be seen that the confinement loss of PCF decreases with the increase of the thickness of the gold film. At the same time, when the refractive index of the measured liquid is *n* = 1.33, the corresponding resonance wavelengths are 570, 586 and 597 nm, respectively. When the refractive index *n* = 1.3402, the resonance wavelengths are 580, 600 and 610 nm, respectively. According to the spectral sensitivity equation, the spectral sensitivity of the sensor increases from about 1000 nm/RIU to 1400 nm/RIU and then decreases to 1300 nm/RIU. Therefore, we find that increasing the thickness of the gold film can improve the spectral sensitivity of the sensor.

However, the loss peak at 35 nm is sharper than that at 40 nm. Considering the full width at half maximum (FWHM) and sensitivity, the following simulation parameters of the gold film thickness are set to 35 nm. According to our assumption, graphene can be coated on a 35 nm thick gold film to achieve the best results. However, there may be an interaction between the gold film and graphene, which results in that the thickness of the gold film does not obtain the maximum sensitivity. Therefore, the data we get through simulation is not necessarily the best.

As mentioned earlier, graphene as a plasma material can enhance the evanescent field. If graphene is coated on the gold film, the resonance wavelength position can be adjusted by enhancing SPR, and finally, the sensitivity of the sensor can be improved. Because the fabrication of the sensor failed to transfer graphene with fixed layers, the effects of graphene films with different thickness on the performance of the sensor were simulated. As can be seen from Figure 3, the coating of graphene on the gold film causes the resonance wavelength of SPR to move towards infrared. Compared with the sensor with the same structural parameters without graphene coating, when the thickness of the graphene dg = 1.7 nm, the resonance wavelength shifts from 586 nm to 607 nm. Moreover, the increase of graphene thickness leads to the increase of PCF confinement loss and the right shift of resonance wavelength. The numerical results show that the confinement losses corresponding to graphene thickness of 1.02 nm and 1.7 nm are 35 dB/cm and 39 dB/cm, respectively, and the resonance wavelength is redshifted from 598 nm to 607 nm.

In order to show whether graphene can improve the performance of the sensor, we simulated the confinement loss of PCF in the range of 1.33–1.3688. Figure 4a shows that the resonance wavelength shifts from 586 nm to 652 nm when the sensor is coated with only 35 nm gold film. Its average spectral sensitivity is about 1700 nm/RIU. In Figure 4b, the resonance wavelength shifted from 607 to 686 nm after graphene was coated on the gold film. Its average spectral sensitivity is about 2040 nm/RIU. Compared with the former, the presence of graphene enhances the SPR of the gold film of the sensor and improves the detection performance of the sensor, which is of research significance.

If the liquid refractive index sensor is used to measure the actual data, it should pay great attention to the accuracy of the sensor measurement and the reliability of the measurement results at the beginning of the design.

The black curve in Figure 5 shows that graphene is not coated, and the red curve shows that graphene is coated on optical fibers. By fitting the relationship between the resonant wavelength and the refractive index of the sensor, we find that it has good linearity. Moreover, the existence of graphene can improve the determination coefficient of fitting straight line and make R^2^ closer to 1. This proves that the linear regression equation has reference value.

After completing the simulation, we made the PCF included in the sensor system. All parameters are consistent with the simulation settings as far as possible. After that, we successfully deposited the gold film with a thickness of 35 nm in the depression of PCF by electron beam evaporation. Electron beam evaporation is a kind of physical vapor deposition. It evaporates gold into the film by accelerating electron bombardment of the pure gold source. It can accurately achieve high purity and high precision metal deposition by using electromagnetic field. Electron beam evaporator can control the thickness of the gold film. As the gold film we steamed is very thin, the thickness may be limited by the sensitivity of the machine sensor with an error of 1–2 nm. Figure 6 shows the evaporated PCF fragment. Hundreds of PCFs meeting the requirements of metal film thickness can be obtained by one evaporation. The quality of the metal film is guaranteed, and the manufacturing cost is reduced.

Due to the limitation of experimental equipment, we can only observe the side of PCF through 500 times magnification of the optical microscope. In Figure 7, we observe the side of a PCF microscope through transmitted and reflected light. The gold film has high smoothness and can be used in experiments.

Afterwards, we treated graphene ethanol dispersions with 1–10 layers and a concentration of 1 mg/mL with ultrasound in order to eliminate agglomeration. The solution was purchased by a commercial company. Considering the feasibility of different transfer methods of graphene, we decided to deposit graphene in defects of PCF by multiple pull-up and dry PCF. The number of pull-ups is 20. Of course, this method is not perfect. It is easy to bend the optical fibers, which may lead to uneven distribution and different thickness of graphene films. Therefore, the experimental results may be affected and need further improvement. However, according to our simulation results of graphene thickness, we can find that the thicker the graphene film, the more the resonance wavelength shifts. We believe that the thickness of graphene film can be increased by increasing the number of pull-ups. But for a single sensor, it has its own graphene film thickness and measurement standards. This must be taken into account.

We have built a detection system, as shown in Figure 8, for the experiment of the sensor. Then we fused two MMFs at the two ends of a section of PCF which is 1 cm long as fiber optic patch cord to connect the laser and the spectrometer. The fusion of optical fibers will cause loss, but this does not affect the sensitivity of the sensor. Due to the loss does not affect the position of the resonance peak. By improving the parameters of the optical fiber fusing machine, we can realize the non-collapse of the air hole in PCF. The diameter of multimode optical fibers is 125 μm, and the core diameter is 62.5 μm. Light travels from the light source into the core of PCF through the multimode fiber optic patch cord. If the light is not successfully coupled into the core of PCF, the resonance peak may not be detected by the spectrometer. SPR is generated by coupling the evanescent field with SPW at a specific wavelength. In the experiment, we only need to immerse part of PCF in the measured liquid marked blue in the figure. In order to allow light to touch the liquid environment, we did not add a cladding outside the PCF. We collect the transmitted light at the output end of the sensor and analyze the refractive index of the liquid according to the change of the transmission spectrum curve and the movement of the trough. It refers to controllable graphene film thickness and better deposition methods.

In order to complete the experiment, we configure different mass fractions of glucose solution. The refractive index of glucose solution measured by refractometer is shown in Table 1. The glucose solution used is all made up of deionized water, and the glucose solution with a mass fraction of 0 represents water.

Finally, we have completed the sensor detection experiments by two methods: Only gold plating and coating graphene after gold plating. The obtained transmission spectrum is filtered and redrawn, as shown in Figure 9.

Figure 9 shows that the experimental and simulation results show the same trend of confinement loss and resonant wavelength. Figure 9a,b show the measured data of sensors without or with graphene coating, respectively. With the increase of the mass fraction of the glucose solution, the confinement loss of the fundamental mode of the optical fiber increases, which can be reflected by the decrease of the transmittance and the red shift of the resonance wavelength. The resonance wavelengths obtained by the sensor without graphene in detecting water and a 25% glucose solution are 589 and 657 nm, with a change of 68 nm. The corresponding resonance wavelengths of the sensor containing graphene were 619 and 701 nm, respectively, with a change of 82 nm. We find that the average spectral sensitivity of the sensor increases from 1900 nm/RIU to 2290 nm/RIU with a graphene coating, which is close to the simulation results. From the experimental point of view, we proved that graphene could enhance SPR and improve the performance of the sensor. It is noteworthy that in our experiments, we find that the sensor without graphene produces a wave trough near 547 nm, while the sensor with graphene also has a wave trough near 555 nm. We think this is due to the selection of the reference spectrum of the spectrometer. Because PCF needs to be evaporated with gold before the sensor is manufactured, the transmission spectrum of the sensor before the coating cannot be obtained, which eventually leads to another trough. However, it has no effect on the detection performance of the sensor and is still sensitive to the refractive index.

Considering that the sensor should be durable, we have tested the graphene-coated sensor repeatedly. It is assumed that if the sensor is stable for a long time, its resonance wavelength should be constant. Therefore, we have experimented on the same gold film sensor coated with graphene. In this test, we measure the refractive index of water every 24 h and record its resonance wavelength. PCF is washed and dried after each measurement. And the experiment is repeated five times. The red curve represents the test curve, and the green curve represents the test completion curve. As we can see from the red curve in Figure 10, the transmittance increases gradually with each measurement. But the position of the resonance wavelength is not shifted. The final data in Figure 10 show that the resonance wavelength and sensitivity of the sensor to water remain unchanged, while ignoring the effect of light source instability on transmittance. Based on this result, we believe that the sensor has good durability and can be reused for a long time.

Similarly, we also fitted the data of the two groups, as shown in Figure 11. The black curve indicates that no graphene is coated, and the red curve indicates that the optical fibers are coated with graphene. Graphene-free sensor has very high linearity, which is beyond our expectation. High linearity can bring higher measurement accuracy and reliability. This proves that the design of the sensor is reasonable. The linearity of sensors with graphene is slightly worse, which may be due to the uneven distribution of graphene films. If the quality of graphene can be improved, such as the controllable thickness of graphene film and better deposition method, the performance of the sensor may be improved. These methods will increase the linearity of the sensor and make the experiment and simulation more consistent.

## 4. Conclusions

An SPR liquid refractive index sensor based on PCF is proposed in this paper. Firstly, SPR is generated by gold film, and a high linearity sensor with an average spectral sensitivity of 1900 nm/RIU is obtained in the range of liquid refractive index from 1.3330 to 1.3688. After that, we coated graphene on the gold film and obtained a graphene optical fiber sensor with an average spectral sensitivity of 2290 nm/RIU. The sensitivity of 390 nm/RIU is improved on the basis of the original sensor. In this paper, the thickness of the gold film is precisely controlled by electron beam evaporation, and the spectral sensitivity of the sensor is successfully improved by coating graphene layer. The sensor designed has low cost and good reusability. It has potential practical application value.

## Figures and Tables

**Figure 1 sensors-19-03666-f001:**
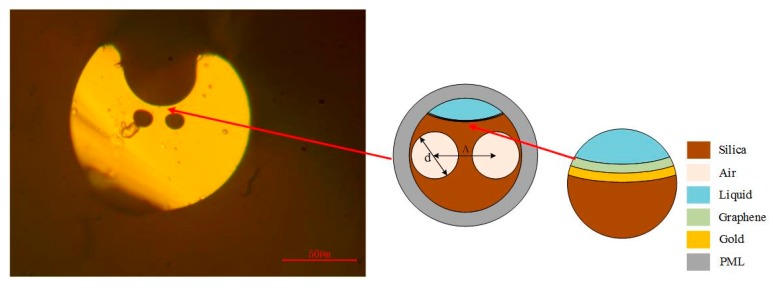
Structure of the sensor.

**Figure 2 sensors-19-03666-f002:**
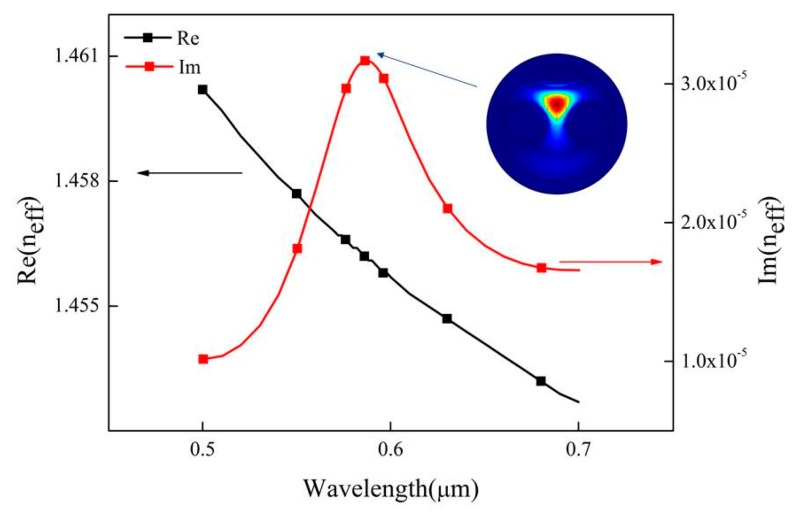
The real and imaginary parts of the effective refractive index of the fundamental mode change with wavelength. Conditions: *n* = 1.33.

**Figure 3 sensors-19-03666-f003:**
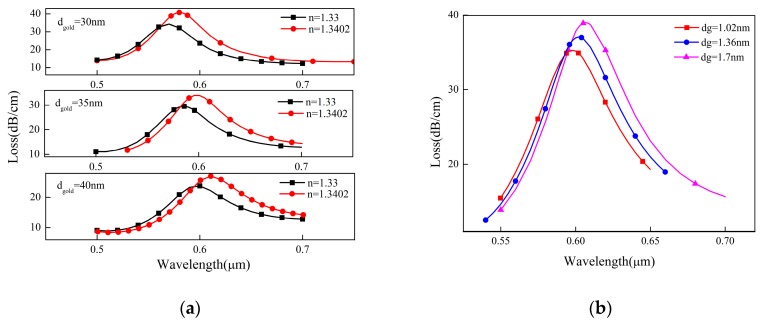
The influence of the gold film and graphene thickness. (**a**) The effect of the gold film thickness on photonic crystal fiber (PCF) confinement loss; (**b**) The effect of graphene thickness on PCF confinement loss.

**Figure 4 sensors-19-03666-f004:**
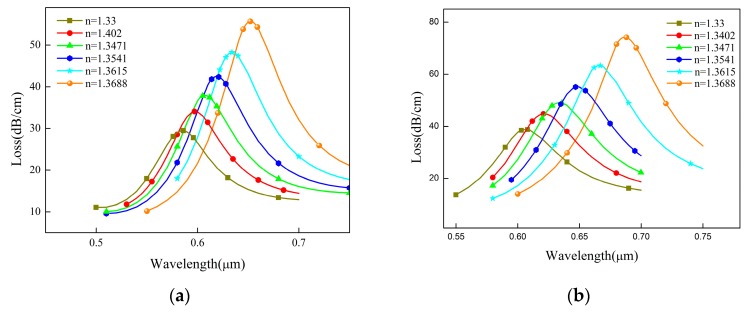
The comparison of the gold film with or without graphene coating. (**a**) Analysis of the confinement loss of PCF without graphene; (**b**) Analysis of the confinement loss of graphene-coated PCF.

**Figure 5 sensors-19-03666-f005:**
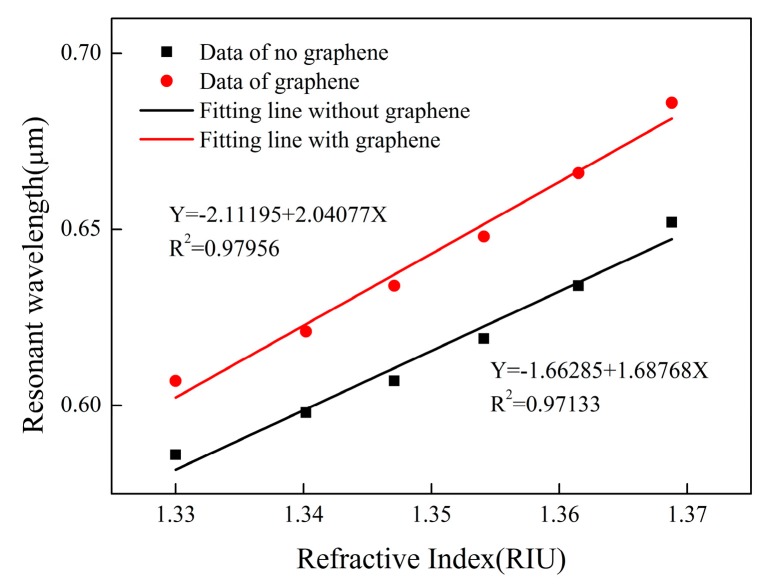
Comparison of sensor fitted lines with or without graphene.

**Figure 6 sensors-19-03666-f006:**
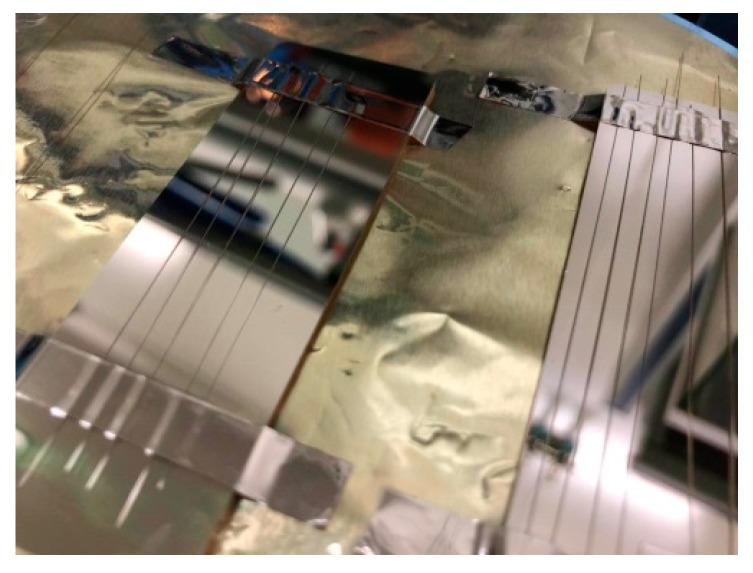
Gold-plated optical fiber.

**Figure 7 sensors-19-03666-f007:**
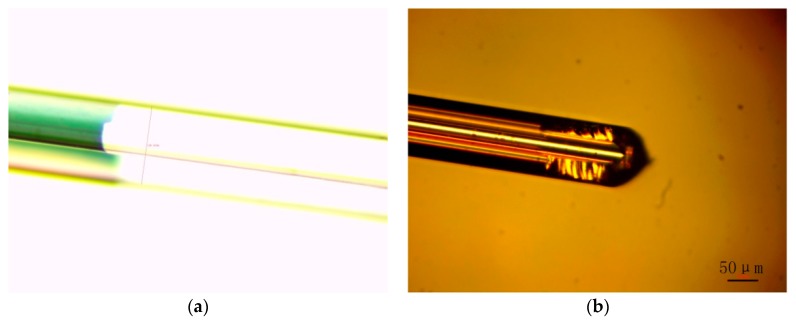
Microscopic observation of PCF. (**a**) Transmitted light image; (**b**) Reflected light image.

**Figure 8 sensors-19-03666-f008:**
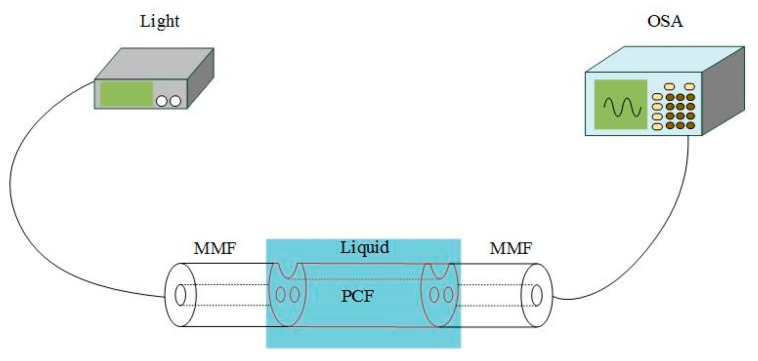
The sensor system.

**Figure 9 sensors-19-03666-f009:**
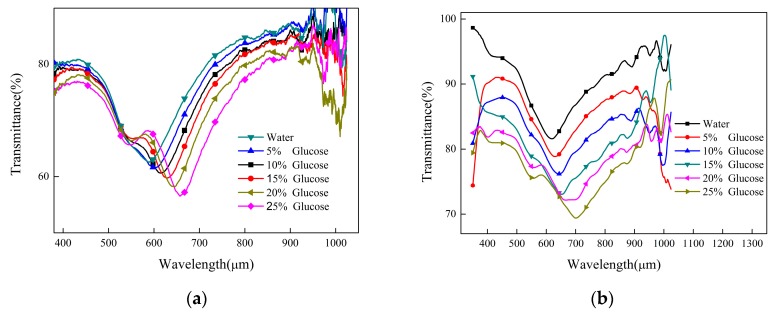
Comparison of experimental transmittance. (**a**) The gold film was not coated with graphene; (**b**) The gold film is coated with graphene.

**Figure 10 sensors-19-03666-f010:**
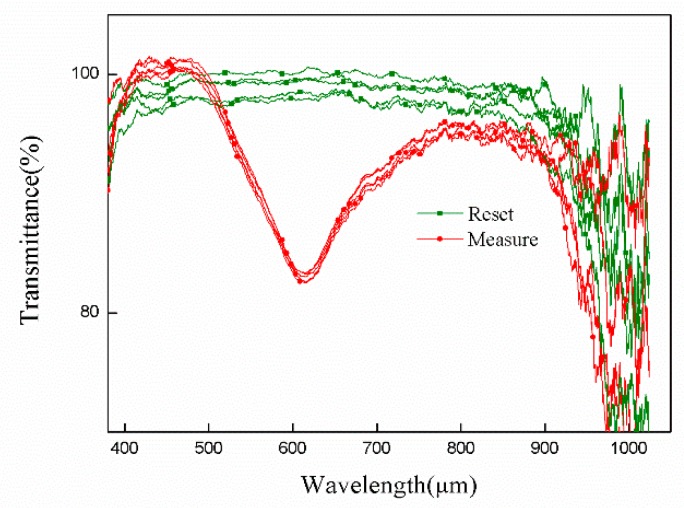
The measurement and recovery curves obtained by repeated testing of the sensors.

**Figure 11 sensors-19-03666-f011:**
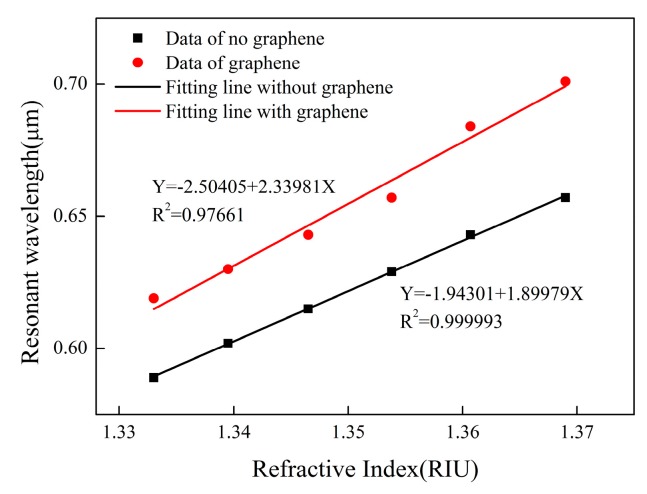
Fitting the experimental data of sensors with or without graphene.

**Table 1 sensors-19-03666-t001:** Relationship between the mass fraction of glucose solution and refractive index.

Mass Fraction (%)	0	5	10	15	20	25
refractive index	1.3330	1.3402	1.3471	1.3541	1.3615	1.3688

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
