# Peer review of "Graphene-Enhanced Surface Plasmon Resonance Liquid Refractive Index Sensor Based on Photonic Crystal Fiber"

_sensors, 2019, doi:10.3390/s19173666_

Round 1

Reviewer 1 Report

The report by Li et al. (Graphene-enhanced Surface Plasmon Resonance Liquid Refractive Index Sensor Based on Photonic Crystal Fiber) reports the surface plasmon resonance (SPR) liquid refractive index sensor based on photonic crystal fiber (PCF). For this, firstly, they coated PCF surface with and gold and graphene, consecutivley. It was shown thatspectral sensitivity of the sensor is successfully improved by coating graphene layer.

The manuscript is well-written and presents some valuable data.  I propose the publishment of this work after consideration of some major points given below.

There is no data showing the presence of the graphene layer onto the PCF. SEM and Raman data might be helful fort he clearification.

In Figure 9, what does a and b stand for? Please define it in the figure caption.

Line 175 It  must be Figure 9 not 10.

Figure 10 must be mentioned in Line 194-198 and also there is no data showing how the durability test was performed. Please add some statements for the clearification.

Author Response

Authors' Responses to Reviewer

Dear Reviewer,

We appreciate the editor and the referees’ insightful and valuable comments. We have carefully revised the manuscript according to the suggestions and comments. Detailed point to point responses and changes are summarized as below. The revisions in the manuscript have been marked in red text. Special thanks to you for your good comments.

Reviewer: 1
Comments and Suggestions for Authors
The report by Li et al. (Graphene-enhanced Surface Plasmon Resonance Liquid Refractive Index Sensor Based on Photonic Crystal Fiber) reports the surface plasmon resonance (SPR) liquid refractive index sensor based on photonic crystal fiber (PCF). For this, firstly, they coated PCF surface with and gold and graphene, consecutivley. It was shown thatspectral sensitivity of the sensor is successfully improved by coating graphene layer.

The manuscript is well-written and presents some valuable data.  I propose the publishment of this work after consideration of some major points given below.

There is no data showing the presence of the graphene layer onto the PCF. SEM and Raman data might be helpful for the clarification.

In Figure 9, what does a and b stand for? Please define it in the figure caption.

Line 175 It  must be Figure 9 not 10.

Figure 10 must be mentioned in Line 194-198 and also there is no data showing how the durability test was performed. Please add some statements for the clarification.

Thank you for your comments and suggestions.

There is no data showing the presence of the graphene layer onto the PCF. SEM and Raman data might be helpful for the clarification.
Response: The absence of SEM and Raman data is due to the lack of such equipment in the laboratory. The conclusion that graphene enhances SPR can only be proved temporarily by data.

In Figure 9, what does a and b stand for? Please define it in the figure caption.
Response: It was our negligence. Thank you for your correction. Figure 9 shows a comparison of transmission spectra. We want to compare the effect of graphene on the sensor. It is reinterpreted in line 231.

Line 175.It must be Figure 9 not 10.
Response: We found and corrected it.

Figure 10 must be mentioned in Line 194-198 and also there is no data showing how the durability test was performed. Please add some statements for the clearification.
Response: We reinterpret Figure 10. It will appear in lines 253-263.

Reviewer 2 Report

Authors report on liquid refractive index sensing by using a fiber structure based on the use of a photonic crystal fiber. The surrounding refractive index is measured by exploiting the surface Plasmon resonance phenomenon produced by the interface formed by a gold film deposited onto the fiber. The authors claim that the RI sensitivity is improved when a graphene coating is placed on the surface of the gold film. The use of fiber structures based on PCF and gold film coatings for liquid RI sensing has been proved before. In addition, the sensibility enhancement by graphene on metal coated fiber RI sensors has also been demonstrated. However, the investigation exhibit novelty and significance combining the fiber structure, the deposition technique and the use of graphene for sensitivity enhancement in PCF.  In my opinion, the investigation reported can be accepted for its possible publication in the journal after the authors address the following concerns and comments:

The novelty, significance and importance of the investigation compared with previous reported works must be clearly shown in the introduction section. Details of the photonic crystal fiber must be included, i.e. commercial or homemade fiber? How the he fiber notch was achieved (micro-machined, chemical attack, etc) or the fiber was designed in that way in the fabrication process? Is an unclad fiber? Since thickness of coating is a very important parameter, how the thickness of the deposited gold film was estimated? How the “quality of the metal film is guaranteed”? How the composition of the gold film was determined? E-beam evaporation with some conditions exhibit poor deposition uniformity. Details on deposition process and characterization must be provided. As you mentioned the graphene deposition process used is random and causes inhomogeneity of the graphene film. In the multiple pull-up and dry treatment, how do you determined the optimal treatment times (layers) for the proposed application? And how do you corroborate the graphene film was successfully deposited and its thickness? How the 1 mg/ml concentration of graphene ethanol was determined? In the experimental setup, what are the characteristics of the MMF sections? What is the influence of the core mismatch between spliced fibers? What is the influence of the collapse regions due to the splicing of PCF with air holes? In this regard, how large are the insertion losses of the fiber device? Because figure 10 shows similar confinement loss between simulated and experimental results, but until I understand the simulation does not include the MMF sections or estimated collapse regions losses. The authors claim, “If the quality of graphene can be improved, the performance of the sensor may be improved”. What characteristics of graphene layers the authors assume must be improved (thickness, quality of deposition, concentration, etc.) and why?

The presence of a graphene layer was simulated as a wavelength shift of the resonance peak toward longer wavelengths and a resonance located at longer wavelength exhibit greater sensitivity to RI variations. In this regard, the authors attribute this sensing performance to the addition of a graphene layer, however the wavelength shift can be achived with a thicker material coating (in this case gold), the use of different materials or layered deposition, etc. In this sense, the authors must discuss why a graphene coating is better than traditional methods to induce a wavelength shift of the SPR absorption notch towead longer wavelengths.

Author Response

Authors' Responses to Reviewer

Dear Reviewer,

We appreciate the editor and the referees’ insightful and valuable comments. We have carefully revised the manuscript according to the suggestions and comments. Detailed point to point responses and changes are summarized as below. The revisions in the manuscript have been marked in red text. Special thanks to you for your good comments.

Reviewer: 2

Comments and Suggestions for Authors

Authors report on liquid refractive index sensing by using a fiber structure based on the use of a photonic crystal fiber. The surrounding refractive index is measured by exploiting the surface Plasmon resonance phenomenon produced by the interface formed by a gold film deposited onto the fiber. The authors claim that the RI sensitivity is improved when a graphene coating is placed on the surface of the gold film. The use of fiber structures based on PCF and gold film coatings for liquid RI sensing has been proved before. In addition, the sensibility enhancement by graphene on metal coated fiber RI sensors has also been demonstrated. However, the investigation exhibit novelty and significance combining the fiber structure, the deposition technique and the use of graphene for sensitivity enhancement in PCF.  In my opinion, the investigation reported can be accepted for its possible publication in the journal after the authors address the following concerns and comments:

The novelty, significance and importance of the investigation compared with previous reported works must be clearly shown in the introduction section. Details of the photonic crystal fiber must be included, i.e. commercial or homemade fiber? How the he fiber notch was achieved (micro-machined, chemical attack, etc) or the fiber was designed in that way in the fabrication process? Is an unclad fiber? Since thickness of coating is a very important parameter, how the thickness of the deposited gold film was estimated? How the “quality of the metal film is guaranteed”? How the composition of the gold film was determined? E-beam evaporation with some conditions exhibit poor deposition uniformity. Details on deposition process and characterization must be provided. As you mentioned the graphene deposition process used is random and causes inhomogeneity of the graphene film. In the multiple pull-up and dry treatment, how do you determined the optimal treatment times (layers) for the proposed application? And how do you corroborate the graphene film was successfully deposited and its thickness? How the 1 mg/ml concentration of graphene ethanol was determined? In the experimental setup, what are the characteristics of the MMF sections? What is the influence of the core mismatch between spliced fibers? What is the influence of the collapse regions due to the splicing of PCF with air holes? In this regard, how large are the insertion losses of the fiber device? Because figure 10 shows similar confinement loss between simulated and experimental results, but until I understand the simulation does not include the MMF sections or estimated collapse regions losses. The authors claim, “If the quality of graphene can be improved, the performance of the sensor may be improved”. What characteristics of graphene layers the authors assume must be improved (thickness, quality of deposition, concentration, etc.) and why?

The presence of a graphene layer was simulated as a wavelength shift of the resonance peak toward longer wavelengths and a resonance located at longer wavelength exhibit greater sensitivity to RI variations. In this regard, the authors attribute this sensing performance to the addition of a graphene layer, however the wavelength shift can be achived with a thicker material coating (in this case gold), the use of different materials or layered deposition, etc. In this sense, the authors must discuss why a graphene coating is better than traditional methods to induce a wavelength shift of the SPR absorption notch towead longer wavelengths.

Thank you for your comments and suggestions.

The novelty, significance and importance of the investigation compared with previous reported works must be clearly shown in the introduction section.

Response: The introduction in the article is inappropriate. We have revised the introduction, and analyzed and compared the work of previous scholars. Then we discussed the significance of this work.

Details of the photonic crystal fiber must be included, i.e. commercial or homemade fiber? How the he fiber notch was achieved (micro-machined, chemical attack, etc) or the fiber was designed in that way in the fabrication process? Is an unclad fiber?

Response: The PCF is our own optical fiber. It is a bare optical fiber. In order to fabricate this kind of optical fiber, we have fabricated three different size air holes in a glass rod with a diameter of 20 mm by ultrasonic processing. The larger air holes above are cut by a cutting machine to create a notch and the optical fiber is fabricated by a fiber drawing tower. Although the PCF is our self-made optical fiber, we find that Kostecki et al. of Adelaide University had already designed a better optical fiber with the same structure. We made a supplementary explanation at line 67.

Since thickness of coating is a very important parameter, how the thickness of the deposited gold film was estimated?

Response: As you said, metal coating is a very important parameter. If possible, we can measure the thickness of the gold film by atomic force microscopy. SEM and Raman data might be helpful for the clarification. However, the existing experimental equipment does not have this capability.

How the “quality of the metal film is guaranteed”?

Response: The thickness of gold film is mainly controlled by the sensor of the evaporator itself. The error can be controlled within 2 nm. The quality of gold film is related to the placement of optical fibers in the machine. It is necessary to expose the gap of optical fibers through microscope operation before evaporation. This method can obtain a flat gold film.

How the composition of the gold film was determined?

Response: The gold source we use is pure gold, and the environment of the machine is pure, so we can determine the composition of the gold film.

E-beam evaporation with some conditions exhibit poor deposition uniformity. Details on deposition process and characterization must be provided.

Response: Because of the complex internal structure of the electron beam evaporator and the variety of sensors, we only set parameters on the control panel for evaporation. The specific operation is completed by the team of the evaporator.

As you mentioned the graphene deposition process used is random and causes inhomogeneity of the graphene film. In the multiple pull-up and dry treatment, how do you determined the optimal treatment times (layers) for the proposed application?

Response: In fact, we can't determine the best number of times. Because depending on human manual operation, even if the number of times is the same, there will still be different deposition results.

And how do you corroborate the graphene film was successfully deposited and its thickness?

Response: In the whole experiment, we did not add anything that would affect the sensitivity of the sensor. Therefore, the change of spectra can prove the successful deposition of graphene. If possible, SEM and Raman data might be helpful for the clarification. In both ways, we may be able to measure the thickness of graphene.

How the 1 mg/ml concentration of graphene ethanol was determined?

Response: Graphene dispersions are purchased through commercial channels. Ultrasound can eliminate agglomeration and ensure its concentration.

In the experimental setup, what are the characteristics of the MMF sections?

Response: The diameter of multimode optical fibers is 125 μm and the core diameter is 62.5 μm. Light travels from the light source into the core of PCF through the fiber optic patch cord.

What is the influence of the core mismatch between spliced fibers?

Response: If the core is mismatched, light will propagate mostly in the PCF cladding. This prevents us from detecting SPR.

What is the influence of the collapse regions due to the splicing of PCF with air holes?

Response: The collapse of air holes can lead to unrestricted light transmission. Therefore, it is necessary to constantly modify the parameters of optical fiber fusion to ensure the integrity of air holes.

In this regard, how large are the insertion losses of the fiber device? Because figure 10 shows similar confinement loss between simulated and experimental results, but until I understand the simulation does not include the MMF sections or estimated collapse regions losses.

Response: The insertion loss of MMF used in our experiment is 0.11dB. The confinement loss is analyzed because it can be used to infer the position of the resonance peak of SPR.

The authors claim, “If the quality of graphene can be improved, the performance of the sensor may be improved”. What characteristics of graphene layers the authors assume must be improved (thickness, quality of deposition, concentration, etc.) and why?

Response: The linearity of sensors with graphene is slightly worse, which may be due to the uneven distribution of graphene films. If the quality of graphene can be improved, such as controllable thickness of graphene film and better deposition method, the performance of the sensor may be improved. These methods will increase the linearity of the sensor and make the experiment and simulation more consistent.

The presence of a graphene layer was simulated as a wavelength shift of the resonance peak toward longer wavelengths and a resonance located at longer wavelength exhibit greater sensitivity to RI variations. In this regard, the authors attribute this sensing performance to the addition of a graphene layer, however the wavelength shift can be achived with a thicker material coating (in this case gold), the use of different materials or layered deposition, etc. In this sense, the authors must discuss why a graphene coating is better than traditional methods to induce a wavelength shift of the SPR absorption notch towead longer wavelengths.

Response: The sensitivity of the optical fiber SPR sensor can be improved by special treatment of the metal film, such as coating functional materials or specific materials. Since the sensor is approaching the upper limit of its sensitivity when the SPR is produced by the metal film, we continue to have difficulty in improving the performance of the sensor. The significance of the functional coating layer is not to replace the metal film, but to increase the performance limit of the sensor.

Reviewer 3 Report

Incorporating graphene interface(s) into various devices is a recent research topic. This paper is concerned with PCF based SPR LRI sensor in particular. The results presented will be informative for plasma bio sensing community. The following should be considered and requested in the text.   

To ease readers’ understanding, clearly identify which the data or curves presented in Figs. 2-5,9-11 refer to simulations or measurements either on each figure or in each figure caption

In most general, the sensitivity S would depend on both the gold platting thickness dgo and graphene coating thickness dgr probably with specific nonlinear dependence on them.. Your design, however, is conducted in a manner that graphene coating thickness dgr is optimized after S is maximized for dgr =0 (for bare gold film); in other words, your design  assumed linear dependence of S on dgo and dgr. Reasonable explanation or at least some comment about your assumption (model) is needed.

Author Response

Authors' Responses to Reviewer

Dear Reviewer,

We appreciate the editor and the referees’ insightful and valuable comments. We have carefully revised the manuscript according to the suggestions and comments. Detailed point to point responses and changes are summarized as below. The revisions in the manuscript have been marked in red text. Special thanks to you for your good comments.

 Reviewer: 3

Comments and Suggestions for Authors

Incorporating graphene interface(s) into various devices is a recent research topic. This paper is concerned with PCF based SPR LRI sensor in particular. The results presented will be informative for plasma bio sensing community. The following should be considered and requested in the text.  

To ease readers’ understanding, clearly identify which the data or curves presented in Figs. 2-5,9-11 refer to simulations or measurements either on each figure or in each figure caption.

In most general, the sensitivity S would depend on both the gold platting thickness dgo and graphene coating thickness dgr probably with specific nonlinear dependence on them.. Your design, however, is conducted in a manner that graphene coating thickness dgr is optimized after S is maximized for dgr =0 (for bare gold film); in other words, your design  assumed linear dependence of S on dgo and dgr. Reasonable explanation or at least some comment about your assumption (model) is needed.

Thank you for your comments and suggestions.

To ease readers’ understanding, clearly identify which the data or curves presented in Figs. 2-5,9-11 refer to simulations or measurements either on each figure or in each figure caption.
Response: Thank you for your correction. We have reinterpreted these figures.

In most general, the sensitivity S would depend on both the gold platting thickness dgo and graphene coating thickness dgr probably with specific nonlinear dependence on them.. Your design, however, is conducted in a manner that graphene coating thickness dgr is optimized after S is maximized for dgr =0 (for bare gold film); in other words, your design assumed linear dependence of S on dgo and dgr. Reasonable explanation or at least some comment about your assumption (model) is needed.
Response: Based on your suggestion, we have explained it in line 139. According to our assumption, graphene can be coated on a 35 nm thick gold film to achieve the best results. However, there may be interaction between the gold film and graphene, which results in that the thickness of the gold film does not obtain the maximum sensitivity. Therefore, the data we get through simulation is not necessarily the best.

Author Response

Authors' Responses to Reviewer

Dear Reviewer,

We appreciate the editor and the referees’ insightful and valuable comments. We have carefully revised the manuscript according to the suggestions and comments. Detailed point to point responses and changes are summarized as below. The revisions in the manuscript have been marked in red text. Special thanks to you for your good comments.

 Reviewer: 4

Comments and Suggestions for Authors

The manuscripts shows that using adhesive graphene layer on top of gold layer can improve the PCF SPR sensor performance. This performance improvement technique is well known however, they experimentally showed that which makes the manuscript interesting. Some important issues need address before considering for the acceptance.

Figure 1 shows the proposed sensor where in microscopic image the cladding region is very large in such case where the light in confined I mean which part is working as a core. If possible, I would recommend to include the experimental mode profile. Authors should include the phase matching result which is missing in Fig. 2. The manuscript is written poorly. The figures labelling need to improve significantly. There don’t have proper labelling, even some of the figures are not describe in the text. I would recommend to recheck this very carefully. In Fig. 9, there is no proper indication which one is for only gold and which one for gold-graphene coated? I don’t understand the reason of figure 10. In the text, it is mentioned fig. 10 shows the experimental and simulation results but, I can’t see the proper results. Authors should prepare the results with proper labelling. Introduction section needs to improve significantly. PCF based SPR sensor is a well established research area. Authors should critically reviewed the recently published work. I would recommend to check the below references and compare the sensor performances.

Islam, Md Saiful, et al. "A Hi-Bi Ultra-Sensitive Surface Plasmon Resonance Fiber Sensor." IEEE Access 7 (2019): 79085-79094.

Haider, Firoz, et al. "Propagation Controlled Photonic Crystal Fiber-Based Plasmonic Sensor via Scaled-Down Approach." IEEE Sensors Journal 19.3 (2018): 962-969.

Haider, Firoz, et al. "Highly amplitude-sensitive photonic-crystal-fiber-based plasmonic sensor." JOSA B 35.11 (2018): 2816-2821.

Thank you for your comments and suggestions.

Figure 1 shows the proposed sensor where in microscopic image the cladding region is very large in such case where the light in confined I mean which part is working as a core. If possible, I would recommend to include the experimental mode profile.

Response: The existence of air holes can restrict the propagation of light in the core. The core of the optical fiber refers to the position above the air hole and near the notch. We assume that light will travel in this area. But the fact is that it may leak into the optical fiber cladding. This is mainly due to our failure to control the size of air holes in the process of optical fiber drawing, which reduces the limitation of light.

Authors should include the phase matching result which is missing in Fig. 2.

Response: Thank you for your correction. We added the phase matching result at line 113.

The manuscript is written poorly. The figures labelling need to improve significantly. There don’t have proper labelling, even some of the figures are not describe in the text. I would recommend to recheck this very carefully.

Response: Thank you very much for your seriousness and responsibility. We have completely revised the label in the text.

In Fig. 9, there is no proper indication which one is for only gold and which one for gold-graphene coated?

Response: We also found this problem in the revision and made detailed annotations.

I don’t understand the reason of figure 10. In the text, it is mentioned fig. 10 shows the experimental and simulation results but, I can’t see the proper results. Authors should prepare the results with proper labelling.

Response: The problem in Figure 10 is due to unclear labeling. We have made amendments.

Introduction section needs to improve significantly. PCF based SPR sensor is a well established research area. Authors should critically reviewed the recently published work. I would recommend to check the below references and compare the sensor performances.

Response: The introduction in the article is inappropriate. We have revised the introduction, and analyzed and compared the work of previous scholars. Then we discussed the significance of this work. Thank you for providing us with the latest research results, we will carefully reference.

Round 2

Reviewer 1 Report

I recommend the publication of the revised manuscript. 

Author Response

Thank you very much for your comments.

Reviewer 4 Report

Authors did the corrections accordingly except que. 2. I would recommend to include the phase matching curves for Core-mode, spp-mode. Also, provide the mode profile for coe-mode, spp mode and at the resonant wavelength. At the moment, mode profile is given only at the resonant wavelength. 

Author Response

Thank you very much for your valuable comments. We regret that figure 2 was not properly supplemented in the first revision. Therefore, we re-simulated the model and analyzed each mode. But we have not got the image of SPP mode, so we can not analyze the data of SPP mode alone. The phase matching conditions you mentioned are very important, but our work focuses on whether graphene can improve the performance of the sensor. Maybe it's because we still have problems with the analysis method of the model. We'll continue to study it in detail.